# A strategy to incorporate prior knowledge into correlation network cutoff selection

Elisa Benedetti [1,2], Maja Pučić-Baković [3], Toma Keser[4], Nathalie Gerstner [1,5], Mustafa Büyüközkan[1,2], Tamara Štambuk[3], Maurice H. J. Selman[6], Igor Rudan [7], Ozren Polašek[8,9], Caroline Hayward [10], Hassen Al-Amin [11], Karsten Suhre [12], Gabi Kastenmüller [1], Gordan Lauc [3,4] & Jan Krumsiek[1,2 ✉]

Correlation networks are frequently used to statistically extract biological interactions between omics markers. Network edge selection is typically based on the statistical significance of the correlation coefficients. This procedure, however, is not guaranteed to capture biological mechanisms. We here propose an alternative approach for network reconstruction: a cutoff selection algorithm that maximizes the overlap of the inferred network with available prior knowledge. We first evaluate the approach on IgG glycomics data, for which the biochemical pathway is known and well-characterized. Importantly, even in the case of incomplete or incorrect prior knowledge, the optimal network is close to the true optimum. We then demonstrate the generalizability of the approach with applications to untargeted metabolomics and transcriptomics data. For the transcriptomics case, we demonstrate that the optimized network is superior to statistical networks in systematically retrieving interactions that were not included in the biological reference used for optimization.

[1] Institute of Computational Biology, Helmholtz Center Munich - German Research Center for Environmental Health, 85764 Neuherberg, Germany. [2] Department of Physiology and Biophysics, Weill Cornell Medicine, Institute for Computational Biomedicine, Englander Institute for Precision Medicine, New York, NY 10021, USA. [3] Genos Glycoscience Research Laboratory, 10000 Zagreb, Croatia. [4] Faculty of Pharmacy and Biochemistry, University of Zagreb, 10000 Zagreb, Croatia. [5] Max Planck Institute for Psychiatry, 80804 Munich, Germany. [6] Leiden University Medical Center, 2333 ZA Leiden, The Netherlands. [7] Usher Institute of Population Health Sciences and Informatics, University of Edinburgh, EH8 9AG Edinburgh, UK. [8] University of Split School of Medicine, 21000 Split, Croatia. [9] Gen-info Ltd., 10000 Zagreb, Croatia. [10] Medical Research Council Human Genetics Unit, Institute of Genetics and Molecular Medicine, University of Edinburgh, EH4 2XU Edinburgh, UK. [11] Department of Psychiatry, Weill Cornell Medicine in Qatar, Doha, Qatar. [12] Department of Physiology and Biophysics, Weill Cornell Medical College in Qatar, Education City, Doha, Qatar. ✉email: jak2043@med.cornell.edu

Network inference, i.e., the reconstruction of biological networks from high-throughput data, has become a popular field in systems biology[1–3]. Interactions among biomolecules extracted from the analysis of large data sets can represent known and predict novel biological mechanisms[4,5], in particular enzymatic reactions in molecular pathways[6–8].

Virtually all network inference methodologies require the definition of a parameter that determines which molecular interactions should be included in the network and which should be discarded. The construction of correlation-based networks commonly requires a series of simple steps (Fig. 1a). First, pairwise correlations between variables are estimated from the data, for which a wide variety of methods is available. The next step is to determine which correlation coefficients are statistically different from zero using a hypothesis test, which produces $p$-values associated with each correlation coefficient. These $p$-values are then compared to a given significance level $\alpha$, typically 0.01 or 0.05, with appropriate multiple hypothesis testing correction. Finally, significant correlations can be visualized and further analyzed as a network, where nodes represent the variables in the data set and edges represent significant correlations.

However, this straightforward network inference pipeline has two major pitfalls that are usually overlooked and substantially affect the robustness and reproducibility of correlation-based network inference. First, for most correlation measures, the resulting network will vary substantially depending on the number of observations available in the data set. In general, the bigger the sample size, the lower the $p$-values. This means that with increasing sample size, weaker correlations become significant and the corresponding network becomes denser (Fig. 1b). Second, different multiple testing correction methods (e.g., Bonferroni[9] or Benjamini–Hochberg[10]) have different underlying

assumptions, such as controlling for the familywise error rate versus the false discovery rate (FDR), respectively. However, in practice, the choice of one method over another is usually not scrutinized adequately. Thus, depending on the arbitrary choice of error correction and significance level, one may obtain vastly different networks (Fig. 1b) which are all statistically sound, but that do not necessarily represent relevant underlying biological mechanisms.

We here address the problem of correlation-based network inference from a different perspective. Instead of a statistically driven cutoff selection, we propose to choose the correlation cutoff that produces the correlation network with the highest agreement to a given ground truth (Fig. 1c, d), hereafter referred to as a biological reference. That is, we search for the network that shows the highest overlap with the known underlying biology, thereby avoiding the above-mentioned arbitrarily determined cutoffs for $p$-values.

We postulate that even a coarse, incomplete, or partially incorrect biological reference is suitable for this approach, as long as a sufficient amount of correct biological knowledge is covered. In many cases, the molecular networks regulating the system under study are not fully known, which results in only partial biological reference being available. For example, often only a few of the pathways of the system under study are well-characterized, and for some systems, detailed biochemical information is not available at all. In these cases, we will demonstrate that one can still use the available prior knowledge as a biological reference and obtain a cutoff that is close to the global optimum.

In this paper, we first show that statistical significance selection is indeed substantially influenced by the data set size for common measures of correlation. We then apply the prior knowledge-based cutoff optimization approach to plasma immunoglobulin G

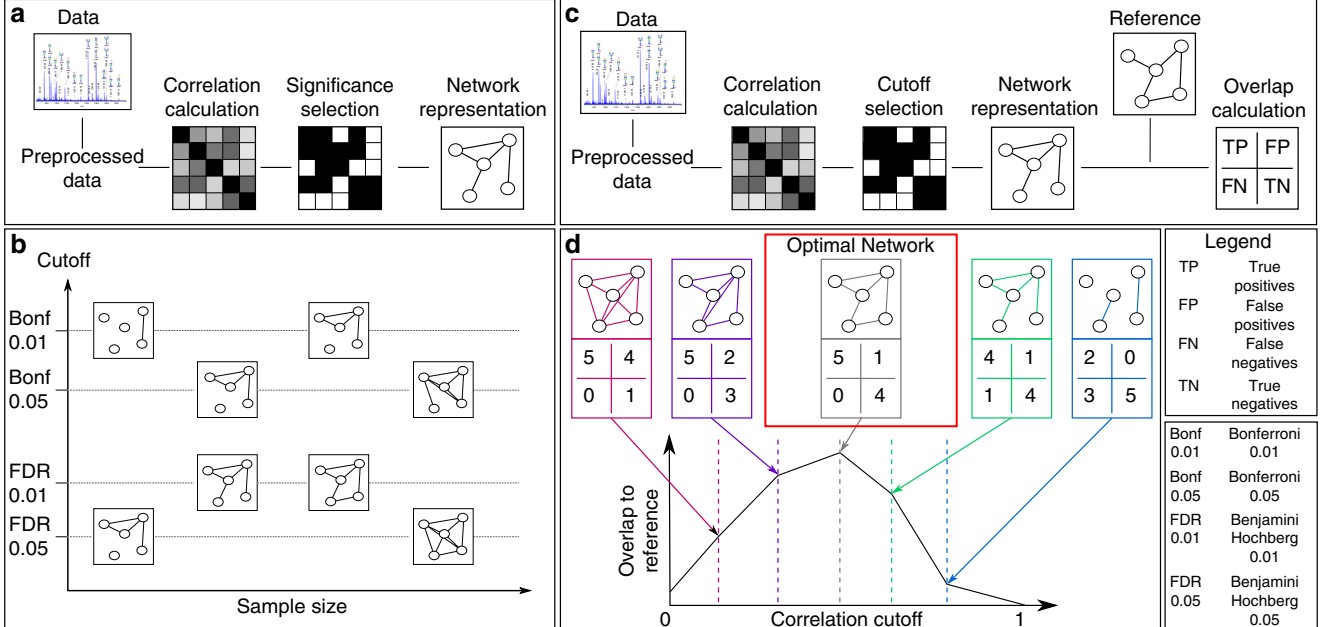

**Fig. 1 Pipeline of network inference and workflow of the paper. a** Typical pipeline of correlation network inference. A correlation matrix is estimated from the preprocessed data. A significant selection step identifies correlations that are statistically different from zero. Significant correlations are commonly visualized as a network. **b** Schematic representation of the dependence of the correlation network on sample size and statistical cutoff. Note that, despite looking substantially different, all resulting networks can be considered statistically correct. **c** Prior knowledge-based network overlap estimation. The correlation network is compared to a prior knowledge network, where the overlap is quantified using true positives (TP), false positives (FP), false negatives (FN), and true negatives (TN). Based on these values, a quality overlap measure between data-driven correlation matrix and biological reference is computed. **d** Prior knowledge-based network inference approach. We discard the $p$-value-based significance selection and instead analyze how the overlap between correlation network and biological reference varies depending on the correlation cutoff. We then define optimal the correlation cutoff at the point where the overlap is maximal.

(IgG) glycomics measurements. In this particular case, we have a well-characterized, supposedly complete biochemical synthesis pathway, which we use as a gold standard biological reference to test our optimization approach. We show that the optimal correlation cutoff is unique and sample size-independent. Moreover, even when the optimization procedure is performed with only a fraction of the original biological reference, the resulting optimum remains the same. We demonstrate the generalizability of the algorithm by applying it to metabolomics data, for which a full detailed prior knowledge is not available. We show that different sources of partial and coarse prior information lead to the same optimal network. Finally, we analyze RNA-sequencing data from The Cancer Genome Atlas (TCGA) to show that the optimized networks are systematically superior to statistical ones in identifying known molecular interactions not included in the optimization procedure. This proves that partial prior knowledge can be exploited to infer data-driven correlation networks that represent true and possibly unknown biological interactions better than regular $p$-value-based networks.

## Results

**Statistical correlation cutoffs depend on a sample size.** For most correlation measures, the larger the sample size, the lower the resulting correlation cutoff at a given significance level. In other words, increasing the number of subjects measured in a study automatically results in a denser correlation network. To quantitatively investigate this effect, we analyzed IgG glycomics measurements from four large Croatian cohorts (Table 1). In the following, the results for one of the four cohorts (Korčula 2013) are shown, while the other three cohorts were used for replication. The discovery data set included 669 samples and 50 glycan structures measured. Data were normalized, log-transformed, and corrected for age and gender prior to analysis.

We subsampled the glycomics data set without replacement to simulate different sample sizes, from 10 to 669 samples. For each subsample, we computed the glycan correlation matrix and applied a 0.01 FDR cutoff using the Benjamini–Hochberg method as an exemplary approach for multiple testing correction. Results would be qualitatively identical with other methods (e.g., Bonferroni) and $\alpha$ levels. We considered two correlation measures commonly used in the field of computational biology: classical pairwise Pearson correlation and partial correlation, which accounts for the presence of confounders (see "Methods" section). We included two different estimators for partial correlation: exact partial correlations obtained from the inversion of the covariance matrix (referred to as parcor), and a shrinkage-based regularization approach, which has been shown to give more stable estimates and still works in data sets with fewer samples than variables (GeneNet[11]).

Note that these two correlation approaches represent substantially different measures of associations: while Pearson correlation quantifies the overall linear association between two variables, partial correlation accounts for the presence of confounders and covariates, therefore removing indirect pairwise associations that only arise due to common mediators. Previous studies have shown that partial correlation selectively represents synthesis steps in metabolic and glycosylation synthesis pathways[8,12], while Pearson correlations are widely used, for example, to infer gene co-expression networks[13]. In this paper, we considered both these measures for comparison.

As expected, for both Pearson correlation and parcor, the significance cutoff, i.e., the smallest still-significant correlation coefficient (in absolute value), decreases with increasing sample size and does not converge even for larger sample sizes (Fig. 2a, red and blue curves, respectively). Interestingly, partial correlations estimated with GeneNet do not show the same behavior, as the statistical correlation cutoff is fairly stable across the considered sample sizes (Fig. 2a, black line). This is also reflected in the total number of edges in the resulting network: While for Pearson correlation and parcor the number of significant coefficients included in the network systematically increases with the sample size, the network estimated with GeneNet maintains a roughly constant number of edges (Fig. 2b). As an example, when considering twice as many samples, from 200 to 400, the GeneNet network remains stable with around 60 edges, while the Pearson correlation network increases by a factor of roughly 1.2 (from 655 to 790), and the parcor network increases by a factor 1.5 (from 95 to 155). Analogous results were obtained in the three replication cohorts (Supplementary Fig. 1). The observed dependence on the sample size can be attributed to how $p$-values are calculated for the various correlation measures. For Pearson and parcor, the degrees of freedom $\kappa$ of the underlying null distribution is proportional to the square root of the sample size, while for GeneNet $\kappa$ is fitted from the regularized partial correlation matrix directly. Since GeneNet partial correlations stabilize quickly even for low sample sizes, $p$-values do not change with increasing sample size $N$. For more details, see Supplementary Methods.

**Reference-based cutoff optimization.** We applied our reference-based network inference approach to IgG glycomics data, for which the pathway of synthesis is well-characterized (Fig. 3a). We have previously shown that edges in a partial correlation network represent a single enzymatic reaction in the IgG glycosylation pathway[8]. In this first step, we tested how our method compares to regular statistical cutoffs.

As a quantitative measure of overlap, we used Fisher's exact test based on the overlap contingency table, which classifies glycan–glycan pairs based on whether an edge between them appears both in the correlation network and in the biological reference (true positives), only in the correlation network (false positives), only in the biological reference (false negatives), or in neither (true negatives). Thus, the higher the overlap between the

**Table 1 Characteristics of the four analyzed glycomics cohorts.**

| | Korčula 2013 Discovery | Korčula 2010 Replication | Split Replication | Vis Replication |
|---|---|---|---|---|
| Number of measured glycans | 50 | 50 | 50 | 50 |
| IgG1 | IgG2 | IgG4 | 20 | 20 | 10 | 20 | 20 | 10 | 20 | 20 | 10 | 20 | 20 | 10 |
| Total number of samples | 695 | 951 | 994 | 780 |
| Males | females | 277 | 418 | 339 | 612 | 390 | 604 | 326 | 454 |
| Samples with no missing values | 669 | 849 | 980 | 729 |
| Unrelated samples | 669 | 504 | 980 | 395 |
| Males | females | 271 | 398 | 156 | 348 | 386 | 594 | 152 | 243 |
| Age range | 18–88 | 18–90 | 18–85 | 18–91 |
| (mean, standard deviation) | (53, 16) | (56, 14) | (50, 14) | (55, 15) |

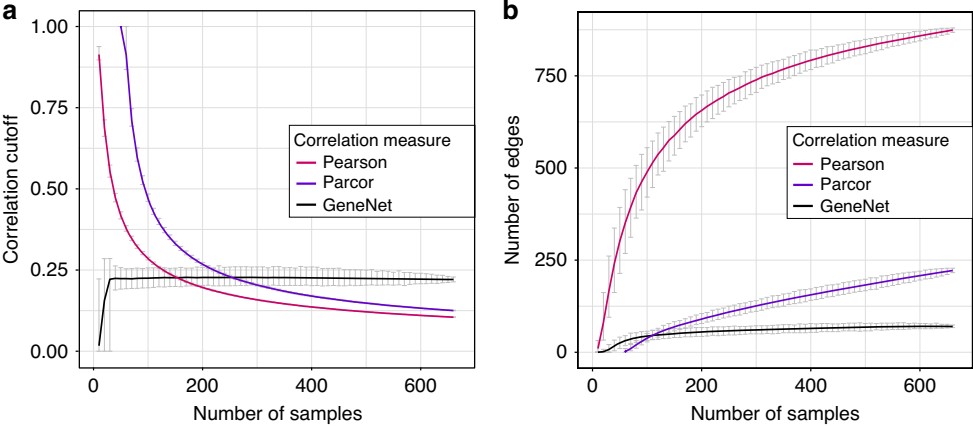

**Fig. 2 Correlation cutoff as a function of the sample size. a** Correlation cutoff (0.01 FDR) as a function of the data set sample size for the three correlation measures considered: Pearson correlation (red), exact partial correlation (purple), shrinkage partial correlation based on GeneNet (black). For each data set size, the original data set was subsampled 1000 times. Curves represent the mean correlation cutoff across the subsampling results for the different correlation measures. Error bars represent the 95% confidence intervals across the subsamplings. **b** Number of edges in the correlation network after applying a 0.01 FDR cutoff as a function of the data set sample size. For each data set size, the original data set was subsampled 1000 times. Curves represent the mean number of edges across the subsampling results for the different correlation measures. Error bars represent the 95% confidence intervals across the subsamplings. Note that for parcor, correlation coefficients can only be estimated for a sample size greater than or equal to the number of variables, in this case, 50.

correlation network and biological reference, the lower this p-value will be (see "Methods" section). The cutoff that produces the maximum overlap to the biological reference is hereafter referred to as the optimal cutoff and the corresponding network as the optimal network. As expected, the regular Pearson correlation performed poorly in comparison to parcor, as it does not account for confounding factors, while GeneNet was the overall best performing method (Fig. 3b). In this case, the optimal GeneNet network yields only a minor improvement over the network obtained with FDR 0.05, which turned out to be the statistical cutoff closest to the optimum. However, the performance of any statistical cutoffs cannot be predicted a priori and depends on the specific case under investigation. The analysis of the replication cohorts showed similar results (Supplementary Fig. 2). This first analysis proves that biological prior knowledge improves the choice of a network cutoff, and that the optimal network is identifiable and unique for all correlation measures considered.

To assess whether the optimal network obtained with our procedure depends on sample size, as statistical cutoffs, we again performed the optimization procedure on subsamples of the original data set (Fig. 3c). For GeneNet, the optimal cutoff turned out to be sample size-independent, as expected. This indicates that, by optimizing the cutoff with our approach even with a relatively small sample size (roughly 160 observations), we still obtain the same optimal network that we would get with a much larger data set (669 observations). Strikingly, even for parcor and Pearson correlation, for which statistical cutoffs showed strong sample size dependence, the optimal cutoff appeared to be sample size-independent over 300 samples (Supplementary Figs. 3 and 4, respectively), although the overall performance was lower than GeneNet. In conclusion, using prior information to optimize the correlation cutoff allowed to infer the same optimal network regardless of the sample size of the data set.

**Incomplete, incorrect, or coarse biological references**. Our optimization approach determines the correlation cutoff at which the data-driven network best represents the biological reference. However, IgG glycan synthesis is a well-characterized process, while in most other practical cases a reference that describes the system in accurate detail is not available. We postulate that even

with an incomplete or partially incorrect biological reference, we will obtain a close-to-optimal network. To this end, we considered the performance obtained from the optimization procedure when comparing the full biological reference with an artificially incomplete, incorrect, or coarse version of it, as described in the following.

Scenario 1: Incomplete biological reference. Since for many biological systems the full biochemical pathway of synthesis is not available, we simulated a case in which only a percentage of the IgG glycosylation pathway is known. To this end, we randomly constructed incomplete pathways by selecting a fraction (10–90% in increments of 10%) of the edges in the IgG glycosylation pathway shown in Fig. 3a. For each percentage, we generated 100 different incomplete pathways and used each of them to optimize the correlation cutoff (Fig. 4a). Obviously, due to the increase in false positives, the fewer edges from the original reference we consider, the lower the overlap to the correlation network becomes. Importantly, however, the optimum is highly conserved across the curves, yielding the same optimal cutoff (0.23) regardless of the amount of prior information available. This means that if we only knew, e.g., 50% of the reactions in the IgG glycosylation pathway shown in Fig. 3a, we would still obtain the identical optimal network as we would by using the full pathway.

Scenario 2: Partially wrong biological reference. In many cases, our understanding of how a biological system works might be partially incorrect. Therefore, we considered the possibility of our reference to include wrong information, i.e., a given number of wrong edges. We simulated an increasing number of edge swaps in the IgG glycosylation pathway until we reached full randomization. For each condition, we generated 100 different pathways and performed the optimization procedure on them (Fig. 4b). Again, while the overall performance decreased as expected, the shape of the curve clearly leads to the same optimal cutoff as the original pathway for up to 20 swaps. This means that even when starting with a substantially incorrect prior, as long as partial truth is contained in the reference network, the optimized network will still produce the same network as the one obtained with the complete biological reference.

Scenario 3: Coarse biological reference. Sometimes no detailed biochemical mechanisms are known, but only general biological properties of the molecules in the data set. For example, we know

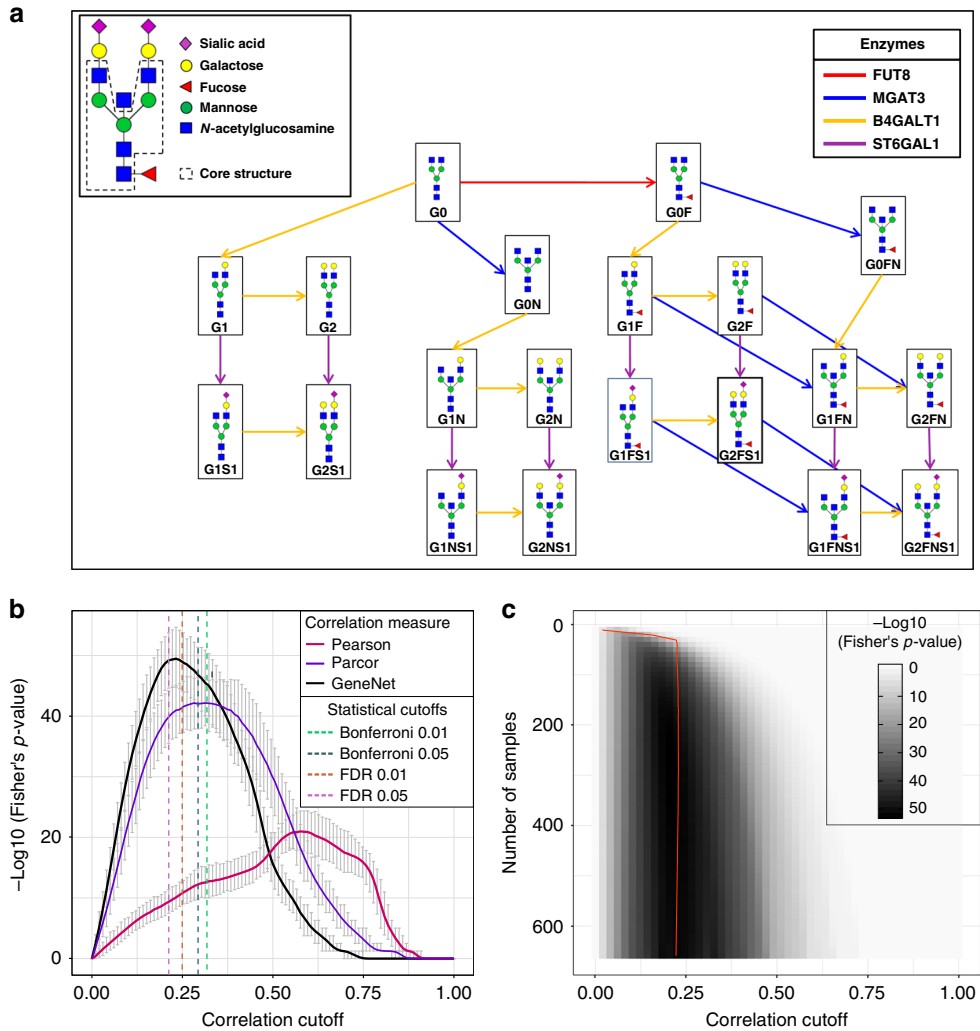

**Fig. 3 Network quality as a function of the correlation cutoff. a** IgG glycan structures and synthesis pathway. The figure was adapted from Benedetti et al.[8] and represents the IgG glycosylation pathway, with nodes representing glycan structures and arrows representing single enzymatic reactions in the synthesis process. **b** Fisher's exact test $p$-values as a function of the correlation cutoff calculated for three correlation estimators: Pearson correlation (pink), exact partial correlation (purple), GeneNet partial correlation (black). For each correlation cutoff, the original data set was bootstrapped 1000 times. Curves represent the mean negative log10 of the two-sided Fisher's exact test $p$-values over the bootstrapping results for the different correlation measures. Error bars represent the 95% confidence intervals of the bootstrapping results. Dashed lines represent the statistical cutoffs for GeneNet on the original data matrix. **c** Mean negative log10 of the two-sided Fisher's exact test $p$-value for partial correlations estimated with GeneNet, as a function of both sample size and correlation cutoff across 1000 bootstrapping. The red line represents the mean of the 0.01 FDR cutoffs across the bootstrapping.

that glycan processing occurs when the sugar chain is already bound to the protein. In our data sets, we have the measurements of three different protein isoforms (IgG1, IgG2, and IgG3 together, and IgG4). Therefore, we can constrain the set of possible biochemical reactions only to glycans pairs within the same IgG isoform (adjacency matrix 1 in Fig. 4c). Moreover, we know that glycosylation enzymes can only add a single monosaccharide at a time during glycan synthesis. Hence, we can further reduce the possible reactions to those between glycan pairs that differ in a single sugar unit (adjacency matrix 2 in Fig. 4c). When comparing the optimization results carried out starting from these biological references to that of the full biochemical pathway (adjacency matrix 3 in Fig. 4c), we observe that, while the overall performance varies, the optimal values are close to each other, thus producing similar networks. Therefore, even when biochemical details are not available for the system under study, other sources of information can be used for the optimization and lead to the same optimum as the complete biological reference.

The three scenarios' results replicated for the other cohorts (Supplementary Figs. 5 and 6).

In conclusion, for various cases of incomplete prior knowledge, our approach still leads to a close-to-globally optimal network.

**Application to metabolomics data.** In order to test whether our approach can be generalized to other data types, we applied the algorithm to untargeted urine metabolomics data set (Table 2). The data set consisted of 95 samples with 1021 measured metabolites and is therefore significantly more complex than the glycomics data set considered so far, which only included 50 variables. Data were normalized, log-transformed, imputed, and corrected for age, gender, and BMI prior to analysis (see "Methods" section).

Since current pathway databases cover only a part of the blood metabolites measured in a typical mass spectrometry-based analysis, we had to rely on partial prior information: (1) Enzymatic reactions connecting the measured metabolites were obtained from the RECON2 database[14]. In addition, as a weak

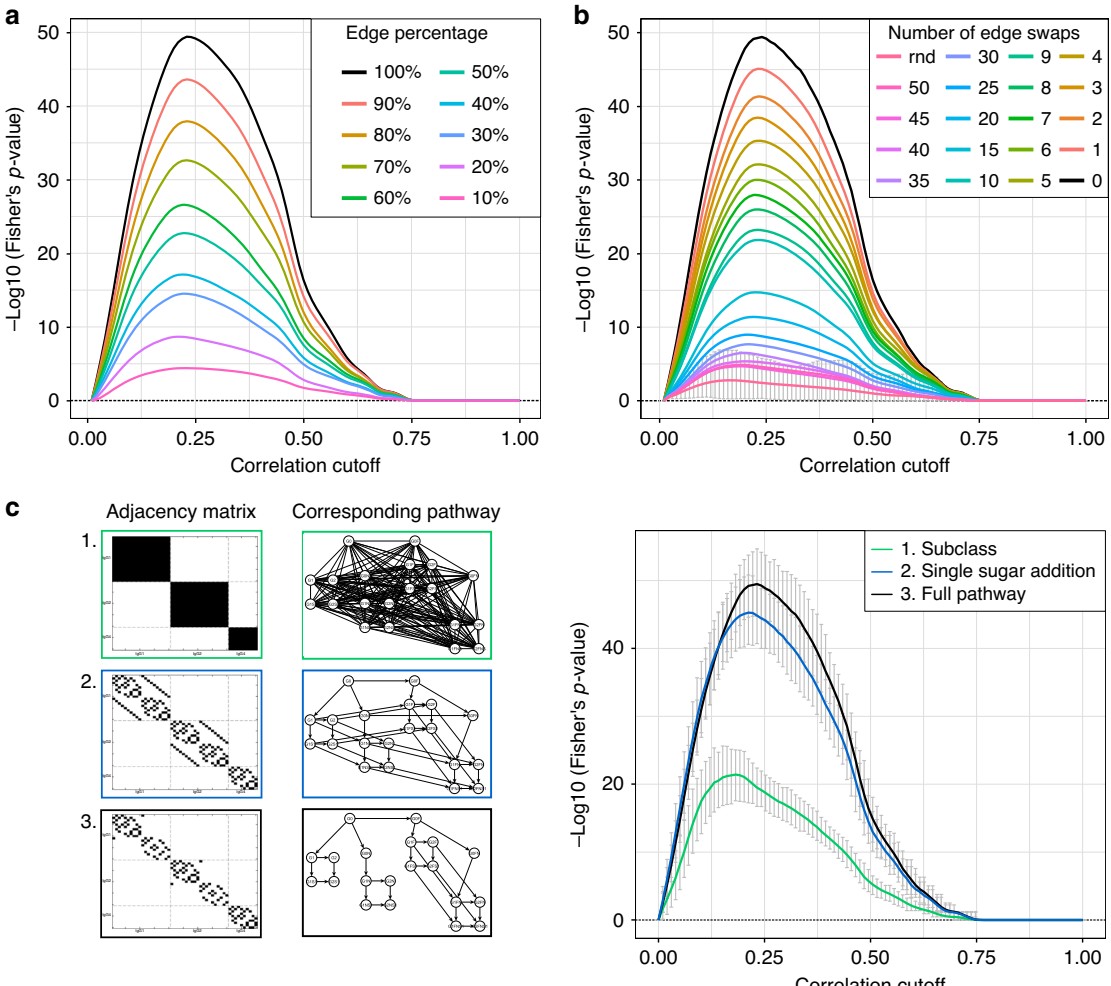

**Fig. 4 Cutoff optimization with partial knowledge. a** Incomplete biological reference. For each percentage, 100 different adjacency matrices were generated by randomly removing edges from the IgG glycosylation pathway. The curves in the figure represent the mean negative log10 of the two-sided Fisher's exact test p-value across 1000 bootstrapping on each adjacency matrix. **b** Incorrect biological reference. Edges in the IgG glycosylation pathway were randomly swapped to simulate incorrect information in the biological reference. For each number of swaps, 100 adjacency matrices were generated and the mean negative log10 of the two-sided Fisher's exact test p-value across those curves and across 1000 bootstrapping are shown as curves in the plot. Here, the red curve represents the mean negative log10 of the two-sided Fisher's exact test p-value across 100 fully randomized adjacency matrices (rnd). The error bars on this curve represent the 95% confidence interval of the bootstrapping. Any signal that falls within these intervals should be regarded as noise. **c** Coarse biological reference. For IgG glycomics data we know that only enzymatic reactions between glycans attached to the same IgG isoform are feasible (adjacency matrix 1, green) and, in addition, that only they can be modified by the addition of one sugar unit at a time (adjacency matrix 2, blue). The black curve corresponds to the optimization performed on the full reference (adjacency matrix 3) for comparison. For each correlation cutoff, the original data set was bootstrapped 1000 times. The curves in the figure represent the mean negative log10 of the two-sided Fisher's exact test p-value across 1000 bootstrapping for the different adjacency matrices. Error bars represent the 95% confidence intervals of the bootstrapping results. In all plots, the black curve corresponds to the optimization performed on the full reference.

**Table 2 Characteristics of the metabolomics cohort.**

| Antipsychotics urine preprocessed | Metabolon validation |
|---|---|
| Number of total metabolites | 1021 |
| Number of known structures | 527 |
| Number of samples | 95 |
| Males \| females | 35 \| 60 |
| Age range | 21–60 |
| (mean, standard deviation) | (36, 8) |
| BMI range | 17–46 |
| (mean, standard deviation) | (28, 5) |

informative prior, we constructed two-block adjacency matrices, allowing interactions among molecules (2) within the same biological pathway (in the following referred to as sub-pathway) or (3) within the same general molecular class (referred to as super-pathway).

We inferred GeneNet-based networks using these three priors as biological references (Fig. 5). Although the absolute performances varied significantly depending on the chosen prior, the maxima were still remarkably close to each other. This means that the corresponding resulting optimal networks will be similar. Similar to the glycomics analysis, the statistical cutoff of FDR 0.05 was found to perform comparably to the optimized cutoff.

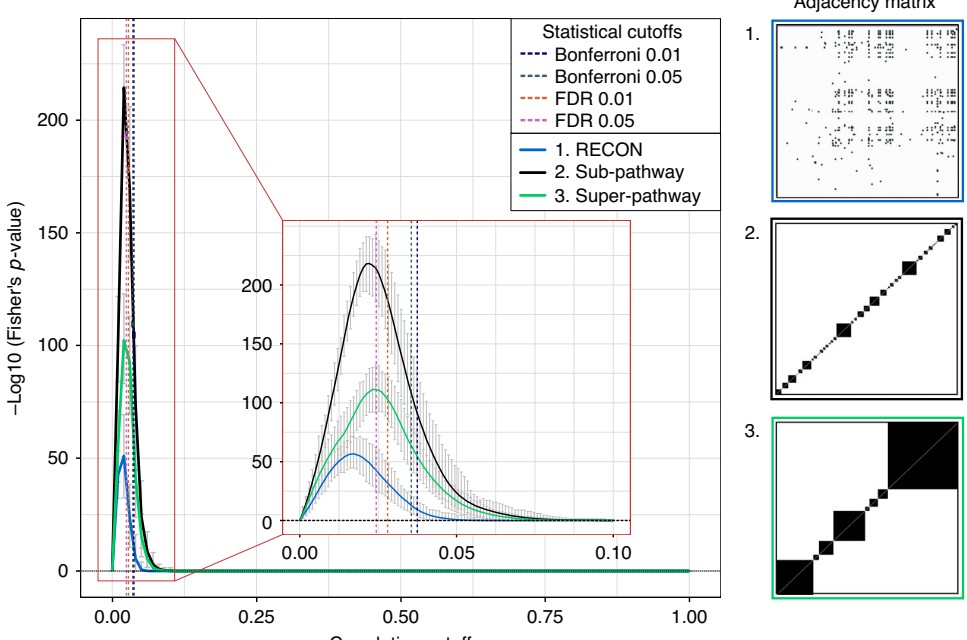

**Fig. 5 Cutoff optimization for metabolomics data.** We used biochemical reactions from the RECON database as partial prior knowledge (adjacency matrix 1, blue), as well as sub- and super-pathway annotations (adjacency matrices 2 and 3, in black and green, respectively). For each correlation cutoff, the original data set was bootstrapped 100 times. Curves represent the mean negative log10 of the two-sided Fisher's exact test *p*-value across the bootstrapping. Error bars represent the 95% confidence intervals of the bootstrapping results. Dashed lines represent the statistical cutoffs for GeneNet on the original data matrix.

For reference, the performances of Pearson and parcor correlation measures can be found in Supplementary Fig. 7.

In conclusion, we demonstrated that our approach can be generalized to metabolomics data, where a full biological reference is unavailable. Partial prior information can be used from different sources and the optima obtained with different priors are highly consistent.

**Application to transcriptomic data.** To prove the ability of our approach to retrieve true biological information, we constructed two entirely independent reference networks and evaluated them using transcriptomics data. These references were based on two popular protein interaction databases: STRING[15] and CORUM[16]. STRING collects information about known and predicted functional associations between proteins, including physical interactions and co-expression data. CORUM is a manually curated database collecting experimentally validated evidence about protein complexes. Notably, these two databases are related since they both contain information on physical interactions between proteins. STRING also contains functional relationships and a higher overall number of protein–protein interactions based on high-throughput data, whereas CORUM focuses on high-quality, manually curated protein complexes. In order to make the two biological priors independent, we removed all interactions contained in the CORUM reference from the STRING reference. Consequently, the two networks did not share any common connections.

We analyzed RNA-sequencing data from The Cancer Genome Atlas[17] (TCGA, Table 3). After preprocessing, the data set included expression measurements of 11,993 genes from 3571 samples across 12 different cancer types. The analysis was conducted separately for 311 different pathways, as defined in the Reactome database[18,19]. For each pathway, we performed a cutoff

| Table 3 Characteristics of the transcriptomics cohort. | |
| --- | --- |
| **TCGA** | **Validation** |
| Original number of transcripts | 16,115 |
| Original number of samples | 3599 |
| Males \| females | 1319 \| 2279 |
| Age range | 18–90 |
| Cancer types | 12 |
| Number of transcripts after preprocessing | 11,993 |
| Number of samples after preprocessing | 3571 |

optimization based on the STRING reference (not containing the CORUM edges), and then computed the overlap of these inferred networks with the CORUM reference. Moreover, to determine the added value of the optimization procedure, we also quantified the overlap of networks calculated with an FDR threshold of 0.05 to both references.

Since we tested 311 pathways in this analysis, we chose a conservative significance threshold for the Fisher's test *p*-value of $0.01/311 = 3.21 \times 10^{-5}$, which yielded 46 pathways with significant overlap to the STRING reference (Fig. 6a and Supplementary Fig. 8). The optimized networks showed a systematically higher overlap with the STRING reference—the reference they were optimized against—compared to the statistical cutoff-based networks, proving that FDR-based network construction would fail in this case. Strikingly, the STRING-optimized networks also outperformed the statistical cutoff networks when compared to the independent CORUM reference (Fig. 6b). An interactive html version of Fig. 6 is provided as a supplement (Supplementary Data 1).

To showcase how these overlap differences between optimized and statistical networks affect the inferred networks, we visualized the partial correlation network obtained with our optimization

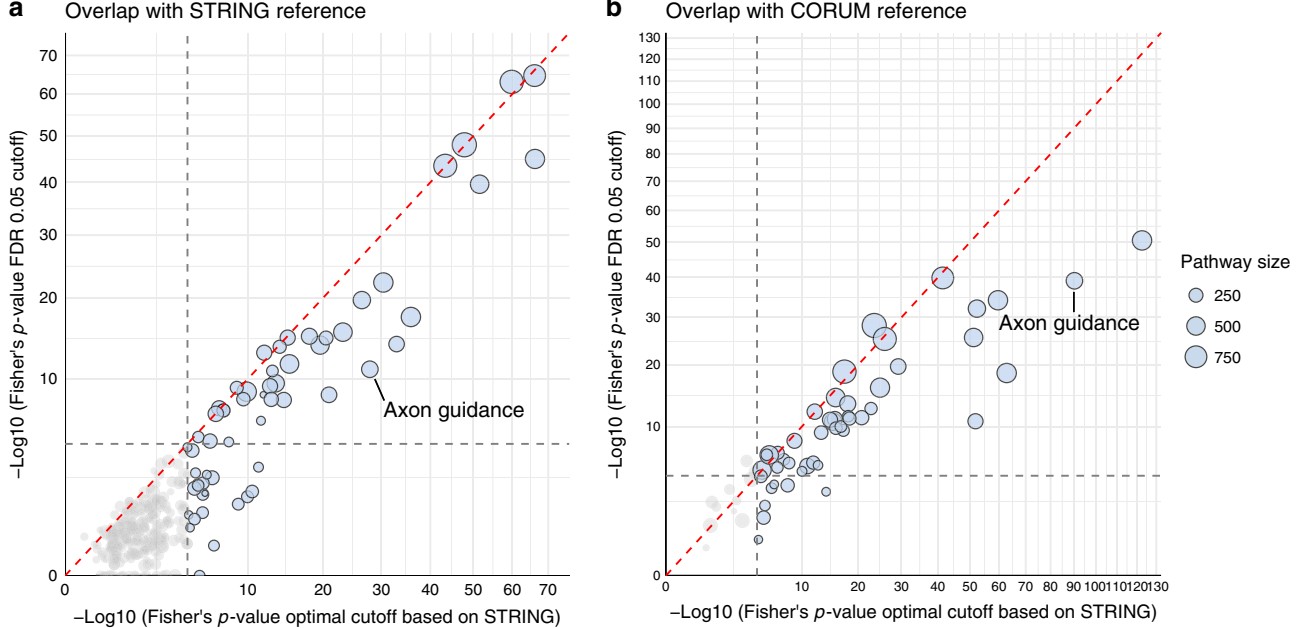

**Fig. 6 Overview of the TCGA transcriptomics data analysis. a** Comparison overlap of statistical (FDR 0.05) and STRING-optimized network with STRING reference. Each dot represents one of the 311 analyzed pathways, where the size of the dot codes for the pathway size. The optimal cutoff was determined with 100 bootstrapping and Fisher's $p$-value was evaluated on the original data set using the bootstrapped optimal cutoff. The gray dashed lines represent the significance threshold of $3.21 \times 10^{-5}$. **b** Comparison of the overlap between the statistical (FDR 0.05) and STRING-optimized network with the CORUM reference. Each dot represents one of the pathways with significant results during optimization (blue dots in **a**). The size of the dot codes for the corresponding pathway size. The gray dashed lines represent the significance threshold of $3.21 \times 10^{-5}$. In both cases, the STRING-optimized networks display a systematic better overlap to the references than the statistically inferred networks.

approach and with a statistical cutoff of 0.05 FDR for the Axon guidance pathway (Fig. 7). Compared to the statistical network, the optimized network is substantially more similar to both the STRING and the CORUM references, displaying a much higher number or edges and thus better reflecting the dense nature of protein–protein interaction networks. A network comparison for all pathways with a significant optimum is provided as an interactive R-Markdown file (see "Code availability" section).

In summary, this analysis showed that our approach can retrieve new biological interactions that were not contained in the reference network used for optimization. Moreover, the analysis demonstrated that the optimized networks are superior to statistical cutoff-based networks.

## Discussion

Correlation network inference often relies on correlation cutoffs based on $p$-values, which are known to be substantially affected by sample size and are subject to an arbitrary choice of significance level and multiple testing correction procedures. We showed that an exception to this general observation is GeneNet[11], which exhibits remarkable robustness to sample size, but is still subject to choice of a proper statistical cutoff.

While several other network inference approaches that do not rely on a $p$-value-based threshold exist, it is worth mentioning that most these methods still require the assignment of a cutoff parameter, for example, the lambda parameter in the graphical Lasso[20], which suffers from the same problem of the $p$-value based cutoff. Other methodologies that do not rely on any cutoff parameter, like, for example, the weighted network approach of WGCNA[21], produce a fully connected network and, although powerful in identifying clusters or modules of co-regulated genes or proteins, are unsuitable to identify single enzymatic steps in synthesis pathways.

The approach presented here overcomes the problem of the cutoff choice by establishing a biological correlation cutoff for network inference. The procedure ranges over the correlation cutoff value until an optimal overlap with a given, possibly incomplete, biological reference is achieved. We benchmarked the approach on LC–ESI-MS IgG glycosylation data from four large Croatian cohorts. For this type of data, the full synthesis pathway has been established and thus served as a gold standard for method evaluation. We showed that for the GeneNet partial correlation method, the resulting optimization curve leads to a well-determined and unique optimum, regardless of sample size and $p$-value cutoffs. The other investigated correlation-based methods performed inferior compared to GeneNet.

The approach was then applied to the more realistic case of partial prior knowledge, i.e., the case where a full, detailed, and correct biological reference is not available. We considered three different scenarios: 1. Only a fraction of the biochemical pathway of synthesis is known. 2. The biochemical pathway contains incorrect information. 3. Only relations between classes of variables are known. In all three cases, we obtained nearly optimal networks despite the reduced biological knowledge that was available. This means that even only marginally informative priors are sufficient to obtain a reasonable approximation of the true network optimum.

We further demonstrated the applicability of the approach to metabolomics and transcriptomics data, for which only partial prior knowledge is available. The three partial biological references used for the metabolomics data, based either on metabolic reactions or molecular annotations, yielded very similar optima. This supports the claim that partial knowledge from different sources, like sub- and super-pathway annotations, can be used to optimize the correlation cutoff.

Interestingly, for both the glycomics and the metabolomics data set, the statistical 0.05 FDR cutoff was close to the optimum.

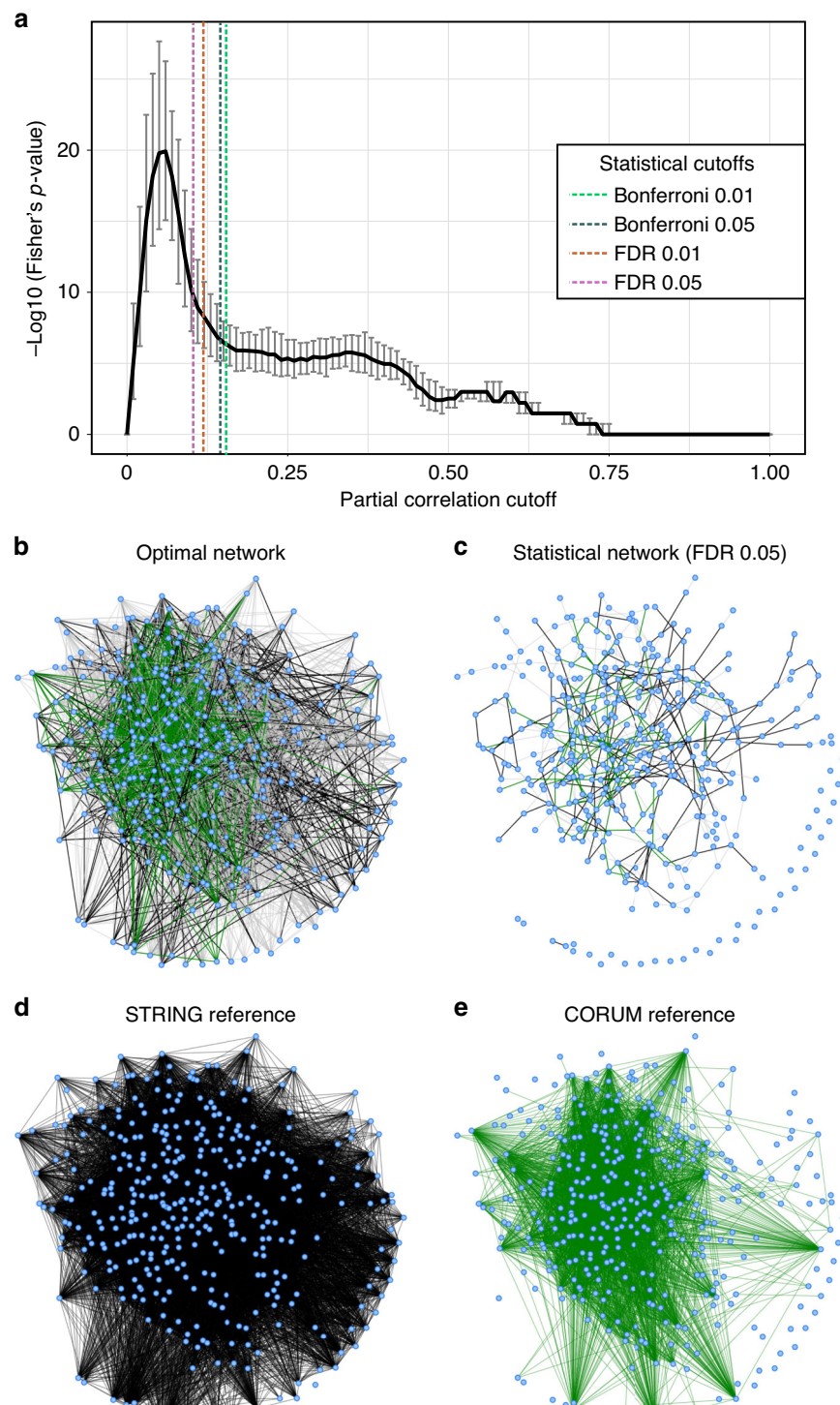

**Fig. 7 TCGA transcriptomics analysis results for the 'Axon guidance' pathway.** This example illustrates how statistical cutoffs can fail to identify a biological optimum. **a** Cutoff optimization. A protein–protein interaction network from STRING was used as a reference. The black curve represents the mean negative log10 of the two-sided Fisher's exact test *p*-value across 100 bootstrapping, and the error bars show the corresponding 95% confidence intervals. Vertical lines indicate the statistical cutoffs computed on the original data matrix. **b** Partial correlation network obtained with our optimization procedure. **c** Partial correlation network obtained with a 0.05 FDR cutoff. **d** Biological reference used for the optimization (PPI network from STRING without CORUM interactions). **e** Biological reference used for validation (protein complexes from CORUM). Black edges represent connections found in the biological reference used for optimization (STRING), green edges represent connections found in the reference used for validation (CORUM), while gray edges indicate connections not included in the prior knowledge.

However, this performance of FDR is coincidental and cannot be generalized to other data sets or data types. This was corroborated by the analysis of transcriptomics data, for which FDR cutoffs were found to significantly overestimate the optimal cutoffs in many cases. This demonstrates that the performance of FDR cannot be known a priori and varies substantially depending on the chosen significance level and data type, and thus does not guarantee an optimal network.

To validate the potential of the optimized networks to infer new biological interactions not included in the biological reference used for the optimization process, we optimized the cutoff of transcriptomics networks based on known protein–protein interactions from the STRING database. We then tested how well the optimized networks represented a different source of information, namely protein complexes from the CORUM database. We constructed the two biological references so that they included complementary information, with no redundancy, and hence were completely independent. Our results show that the STRING-optimized network had a significant overlap with the CORUM reference, and that this overlap was substantially higher than that obtained from statistically inferred networks, meaning that optimizing on partial prior knowledge still allows to correctly infer unknown biology.

The procedure described in this paper requires a quantitative overlap measure to perform cutoff optimization. We chose Fisher's exact test $p$-value as a proxy for the agreement between the calculated correlation network and prior knowledge. It is to be noted that more conventional machine learning measures for classification problems exist. As an example, the popular $F_1$-score[22], does not account for true negatives and was therefore disregarded here. Interestingly, Matthews correlation coefficient[23], another popular measure that uses all values in the contingency table, is mathematically related to the Fisher's $p$-value. Its absolute value is proportional to the square root of the chi-square statistic, which is asymptotically equivalent to that of the Fisher's exact test[24].

Cutoff optimization as presented in this paper is a flexible and generalizable inference strategy. Most other methods that account for prior knowledge integrate the biological reference directly into a specific network inference or regression framework[25–31], for example, by penalizing or enhancing specific edges according to the biological reference. On the contrary, our approach uses prior knowledge as an external reference system to optimize the purely data-driven association matrix. This allows applying the same concept to different association measures, for example, mutual information[32] or other non-linear association quantities, in future studies.

In conclusion, the proposed approach for network inference optimizes the correlation-based cutoff to prior pathway knowledge. The impact of our study lies in the fact that the same optimal network can be obtained when the available prior knowledge is incomplete, partially wrong, or only provides information on the overall relationship across classes of molecules (Fig. 8). Consequently, a network fitted to partial priors can be used to enrich the knowledge with new, previously unknown interactions, or to rectify incorrect links, and can therefore serve as a valuable tool to infer biological interactions when a direct experimental validation is unavailable or unfeasible.

## Methods

**Glycomics data sets.** Plasma IgG glycomics samples from four Croatian cohorts[33] were analyzed (see Table 1 for details). Preprocessed data were downloaded from the figshare repository with the following https://doi.org/10.6084/m9.figshare.5335861[34].

The data sets include measurements of IgG Fc glycopeptides quantified by LC–ESI-MS, which allows the separation of different IgG glycoforms. In Caucasian populations, the tryptic Fc glycopeptides of IgG2 and IgG3 have identical peptide moieties[35,36] and so cannot be separately identified using the profiling method.

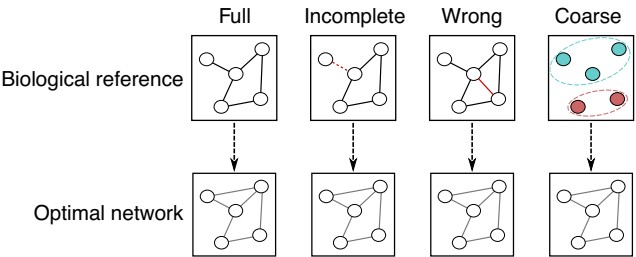

**Fig. 8 Main conclusions of the paper.** We have shown that biochemical pathways can be used to optimize a correlation cutoff to produce the network that best reflects known biological interactions. However, in most cases, a full biological reference is not available. Our approach allows retrieving the same optimal network even when the prior knowledge is incomplete, wrong, or only provides information on the relationship among classes of molecules. The optimized network will still provide the best correlation-based representation of the underlying molecular interactions.

Furthermore, only 10 glycoforms of IgG4 were detectable due to the low abundance of this IgG subclass in human plasma. A detailed description of the experimental procedure used to generate these data can be found in Selman et al.[37] and in Huffman et al.[38] For each IgG subclass, the LC–ESI-MS raw ion counts were normalized using probabilistic quotient normalization, which was originally introduced for metabolomics measurements[39]. The reference sample was calculated as the median value of each glycan abundance across all measured samples. For each sample, a vector of quotients was then obtained by dividing each glycan measure by the corresponding value in the reference sample. The median of these quotients was then used as the sample's dilution factor, and the original sample values were subsequently divided by that value. This procedure was repeated for each sample in the data set. The data were log-transformed and corrected for age and gender prior to statistical analysis.

**Metabolomics data set.** Metabolomic samples were taken from an antipsychotics study conducted in Qatar[40]. Urine samples were analyzed using ultra-high-performance liquid-phase chromatography and gas-chromatography separation, coupled with tandem mass spectrometry by Metabolon, Inc. Data were runday-median scaled, normalized using probabilistic quotient normalization[39], and log-transformed. From the original data matrix, we first excluded metabolites with more than 20% missing values and then samples with more than 10% missing values. Samples with missing covariates were subsequently excluded from the analysis. The filtered data matrix contained 97 samples and 1021 metabolites (527 known structures and 494 unknown), see Table 2. The remaining missing values were imputed with a KNN-based method with variable pre-selection[41]. Data were corrected for age, gender, and BMI prior to analysis.

Approvals for the study were obtained from Hamad Medical Corporation and Weill Cornell Medicine-Qatar Institutional Review Boards. All participants signed written informed consent before joining the study.

**Transcriptomics data set.** RNA-seq data from The Cancer Genome Atlas[17] (TCGA) PANCAN12 study were downloaded from the TCGA Network Research portal (see "Data availability" section) and initially included 3599 samples and 16,115 genes from 12 cancer types: acute myeloid leukemia, bladder urothelial carcinoma, breast invasive carcinoma, colon adenocarcinoma, glioblastoma multiforme, head & neck squamous cell carcinoma, kidney clear cell carcinoma, lung adenocarcinoma, lung squamous cell carcinoma, ovarian serous cystadeno-carcinoma, rectum adenocarcinoma, and uterine corpus endometrioid carcinoma[17], see Table 3. For each cancer type, genes with more than 20% of missing values were excluded. Missing values in the remaining genes were imputed using a KNN-based method with variable pre-selection[41]. Values were corrected for age and gender, and samples without this information were excluded. We only considered genes present in all cancer types after preprocessing and we further corrected for cancer type. The final data set included 3571 samples and 11,993 genes.

**Correlation analysis.** Three measures of correlation were used in this study: (1) Classical Pearson correlation, which represents the linear relation between two variables. (2) Partial correlation, which allows accounting for the presence of confounders or covariates and is calculated as the Pearson correlation coefficient corrected for the presence of all other variables[42]. Analytically, a partial correlation matrix can be obtained by inverting and normalizing the covariance matrix. In this paper, we refer to this technique as parcor. This estimation procedure is efficient but unstable for low sample sizes. (3) A more stable estimate of the partial correlation matrix can be obtained with the GeneNet algorithm[11], where a shrinkage parameter is optimized to correct the covariance matrix prior to inversion. Moreover, the algorithm fits a mixture model to the partial correlation matrix to

compute the p-values[11], which results in a more robust p-value estimation. We attribute to this particular step the observed independence from the data sample size of the partial correlation networks calculated with GeneNet.

Pearson correlation was computed using the stats R package, parcor partial correlations were estimated using the ppcor R package[43] (version 1.1), while GeneNet partial correlation made use of the GeneNet R package[11,44] (version 1.2.15).

For the metabolomics data analysis, partial correlations were corrected for unidentified metabolites. Note that these variables were then excluded from the overlap evaluation during the optimization procedure, which was only based on identifiable metabolites. Statistical cutoffs were based on the full correlation matrix, including unidentified molecules.

**Biological references**. *Glycomics data*: The biological reference reflects the current understanding of the IgG glycosylation pathway, as established in Benedetti et al.[8]. Glycans can be modified by the addition of one monosaccharide at a time, but only selected reactions are enzymatically feasible, as shown in Fig. 3a.

*Metabolomics data*: There is no established complete biochemical pathway to consider as a biological reference for metabolomics data. Known metabolic reactions were imported from the RECON2 database[14] and included in one of the adjacencies. As a more coarse type of biological references, we used sub- and super-pathway annotations provided with the metabolites measurements by Metabolon, Inc., from which adjacency matrices were created by connecting all metabolites within the same sub- or super-pathway, respectively.

*Transcriptomics data*: Pathway annotations were imported from the Reactome database[18,19] through the R package graphite[45,46] (version 1.26.0). We restricted the analysis to pathways containing at least 50 genes and at most 1000 genes and with at least 50% of the genes in the pathway measured in the TCGA data. These constraints led to a total of 311 Reactome pathways being selected. For each pathway, protein–protein interactions were downloaded from the STRING database[47,48] (version 10.5), while protein complexes were taken from the CORUM database[16] (imported from the ganet R package[49], version 2.4.2). CORUM interactions were subsequently removed from the STRING prior, creating two independent references. The resulting modified STRING prior was then used as a biological reference for the optimization, while the CORUM prior was used for validation.

**Overlap estimation**. The overlap between biological reference and correlation network was calculated using Fisher's exact tests, which evaluate whether two categorical variables are statistically independent[50], with low p-values indicating a lack of independence. For the purposes of this analysis, we treated Fisher's p-value as a measure of the overlap between the reference and the calculated correlation network. Lower p-values indicate a better overlap. Therefore, the optimal cutoff for a given correlation is defined as the cutoff at which Fisher's p-value is the lowest. Specifically, all correlation coefficients are first classified in a contingency table, according to significant yes/no, and whether the corresponding variable pair is connected by an edge in the biological reference adjacency matrix (Fig. 3a). The p-value of the two-sided Fisher's exact test is calculated according to the hyper-geometric distribution as:

$$p = \frac{\binom{TP + FP}{TP}\binom{FN + TN}{FN}}{\binom{TP + FP + FN + TN}{TP + FN}} \quad (1)$$

**Reporting summary**. Further information on research design is available in the Nature Research Reporting Summary linked to this article.

## Data availability

Preprocessed glycan concentrations corrected for age and gender were downloaded from the figshare database (URL: https://doi.org/10.6084/m9.figshare.5335861)[34]. Due to patient data protection policies, the metabolomics data cohort cannot be uploaded to a public repository. Data are however available upon request and after compliance with the policies and procedures of Weill Cornell Medicine-Qatar and Qatar National Research Fund for data sharing. Requests can be submitted to Hassen Al-Amin at haa2019@qatar-med.cornell.edu. The PANCAN12 RNA-seq data are freely available for download from the following link: https://xenabrowser.net/datapages/?dataset=TCGA.PANCAN12.sampleMap%2FPanCan12.3602-corrected-v3_syn1715755&host=https%3A%2F%2Flegacy.xenahubs.net&addHub=https%3A%2F%2Flegacy.xenahubs.net&removeHub=https%3A%2F%2Fxena.treehouse.gi.ucsc.edu%3A443[51]. STRING PPI network (version 10.5) was downloaded from https://version-10-5.string-db.org/cgi/download.pl. CORUM protein complexes were taken from the ganet R package (version 2.4.2). Reactome pathways were obtained through the graphite R package (version 1.26.0). Source data are provided with this paper.

## Code availability

R code to reproduce the main results of this paper is publicly available at https://github.com/krumsieklab/CutoffOpt. In order to avoid the long computation time necessary to produce the bootstrapped transcriptomics results presented in this paper, files including precomputed adjacency matrices, preprocessed transcriptomics data and bootstrapping results are made available through figshare (https://doi.org/10.6084/m9.figshare.12646748).

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

## Acknowledgements

The CROATIA_Vis, CROATIA_Korčula, and CROATIA_Split studies were funded by grants from the Medical Research Council (UK), European Commission Framework 6 project EUROSPAN (Contract No. LSHG-CT-2006-018947), FP7 contract BBMRI-LPC (Grant No. 313010), Croatian Science Foundation (Grant 8875), and the Republic of Croatia Ministry of Science, Education and Sports (216-1080315-0302). We would like to acknowledge the staff of several institutions in Croatia that supported the fieldwork, including but not limited to The University of Split and Zagreb Medical Schools, Institute for Anthropological Research in Zagreb, and the Croatian Institute for Public Health. This work was funded in part by grants from the German Federal Ministry of Education and Research (BMBF), by BMBF Grant No. 01ZX1313C (project e:Athero-MED), by the European Commission MIMOmics (contract #305280), IMForFuture (contract #721815) and CarTarDis (contract #602936) grants, and by the European Structural and Investment Funds grant to the Croatian National Centre of Research Excellence in Personalized Healthcare (#KK.01.1.1.01.0010) and Croatian National Centre of Competences in Molecular Diagnostics (#KK.01.2.2.03.0006). The Antipsychotic cohort was funded by Qatar National Research Fund (QNRF) (www.qnrf.org) (Grant No. NPRP 4-268-3-086). K.S. is supported by the Biomedical Research Program at Weill Cornell Medicine in Qatar, a program funded by the Qatar Foundation. The transcriptomics results are based upon data generated by the TCGA Research Network: https://www.cancer.gov/tcga. C.H. is supported by an MRC University Unit Programme Grant MC_UU_00007/10 (QTL in Health and Disease).

## Author contributions

E.B. and J.K. conceived and designed the project. M.P.-B., T.K., T.S., M.J.H.S., I.R., O.P., C.H., H.A.-A., K.S., G.K., and G.L. contributed the data. E.B. and N.G. performed the analyses on the glycomics and metabolomics data, M.B. performed the analysis on the transcriptomics data. E.B. and J.K. wrote the primary manuscript. All authors approved the final manuscript.

## Competing interests

The authors declare the following competing interests: G.L. is a founder and owner of Genos, a private research organization that specializes in high-throughput glycomics. M.P.-B. and T.S. are employees of Genos. All other authors declare no competing interests.
