## [Peer Review File · Nature Communications]

Reviewers' Comments:

Reviewer #1:

Remarks to the Author:

The goal of this manuscript is to develop a method to perform network reconstruction from high-throughput data using prior knowledge instead of an unbiased statistical approach. Examples are presented for IgG glycomics, untargeted metabolomics and TCGA RNA-Seq data. The underlying hypothesis is that 'correct' network synthesis is possible even when missing data are present and when incorrect reference are available. This is an interesting idea that needs experimental validation, particularly for large datasets.

There is inadequate explanation for why correlation cutoff for GeneNet are stable and number of edges constant, based on the statistical cutoff methods alone (Fig. 2). In this regard, the statement 'Optimal GeneNet network (Fig. 3B) outperforms the GeneNet networks obtained with most statistical cutoffs (Fig. 2A)' (gp. 5) is unclear since both approaches reveal similar correlation cutoffs. What is the basis for deciding that one reconstructed network is suboptimal? Perhaps showing the differences in the networks generated using both methods may be helpful.

In Fig. 4, a series of biological reference networks are reconstructed with various levels of error or missing information. During the bootstrapping of the reference network, it is unclear if the incorrect (or missing) information were systematically included (or excluded) in all the runs or if these incorrect/missing data were allowed to be exchanged when generating the adjacency matrices. If exchange is allowed, it is not surprising that the approach can handle incorrect reference information. However, in a real experiment, we would only have one erroneous biological reference. Here, it may not be possible for this approach to identify the 'correct' result in the presence of 'erroneous' reference. The ability of this approach to reveal new information may thus be limited.

The glycomics data is a simple data set with ~20 reactants. The concept outlined by the authors needs to be experimentally verified in more complex systems. Otherwise, all that is being done is to exchange one bias for another.

Overall, this is an interesting approach to calculate optimal correlation cut-off values, but its predictive powers require further validation.

Reviewer #2:

Remarks to the Author:

In this study, Benedetti et al. aimed to develop an algorithm for better correlation-based network inference. Instead of relying on standard statistical cutoffs, their approach selects the cutoff by optimizing the overlap between the inferred and the previously known biological network. It is claimed that the optimal network outperforms networks obtained with statistical cutoffs and is robust to the sample size variation. Overall the paper is well written. The introduction provides good overall coverage of the relevant background literature and description of the problem. However, some sections especially the methodology part needs to be revised/clarified to provide a clearer explanation. Furthermore, it is believed the analysis is quite limited in its scope. These significantly weaken the conclusion of this manuscript and limits its novelty and impact.

Major concerns:

1. There is lack of solid validation of the proposed method. Basically, the study used prior information to select the optimal cutoff based on the overlap between the inferred network edges and the prior network/pathway information (with Fisher's test), but its performance should also be tested in an independent dataset (gold standard).

2. Even with using the same gold standard for selecting and testing the cutoff, the results based on the proposed algorithm are similar with the statistical cutoff (FDR 0.01 or 0.05, Figures 3 and 5). There is only minor improvement over existing statistical methods when applying the algorithm on the IgG glycomics and metabolomics data analysis. For the transcriptomics data, while FDR 0.05 may yield a much sparse network, the problem can be simply overcome by relaxing the FDR cutoff, eg. FDR 0.1.

3. The classical Pearson correlation and the partial correlation correspond to completely different biological concepts in the context of network inference. The manuscript doesn't provide sufficient clarification of both measures, instead, it simply compared them and made a conclusion that the partial correlation using GeneNet is superior compared to others. Detailed explanation should be provided on why this is the case.

4. It should be made clearer how the p-values (before multiple testing correction) were obtained for the Pearson correlation and the partial correlation measure. The different p-value estimation methods also affect the performance.

Response to the reviewers

Reviewers' comments:

Reviewer #1 (Remarks to the Author):

The goal of this manuscript is to develop a method to perform network reconstruction from high-throughput data using prior knowledge instead of an unbiased statistical approach. Examples are presented for IgG glycomics, untargeted metabolomics and TCGA RNA-Seq data. The underlying hypothesis is that 'correct' network synthesis is possible even when missing data are present and when incorrect reference are available. This is an interesting idea that needs experimental validation, particularly for large datasets.

There is inadequate explanation for why correlation cutoff for GeneNet are stable and number of edges constant, based on the statistical cutoff methods alone (Fig. 2). In this regard, the statement 'Optimal GeneNet network (Fig. 3B) outperforms the GeneNet networks obtained with most statistical cutoffs (Fig. 2A)' (gp. 5) is unclear since both approaches reveal similar correlation cutoffs. What is the basis for deciding that one reconstructed network is suboptimal? Perhaps showing the differences in the networks generated using both methods may be helpful.

Correlation measures.

We agree with the reviewer that the stability of GeneNet needs more explanation. We adapted the corresponding result section as follows:

"The observed dependence on the sample size can be attributed to how p-values are calculated for the various correlation measures. For Pearson and parcor, the degrees of freedom κ of the underlying null distribution is proportional to the square root of the sample size, while for GeneNet κ is fitted from the regularized partial correlation matrix directly. Since GeneNet partial correlations stabilize quickly even for low sample sizes, p-values do not change with increasing N. For more details, see Supplementary Methods."

Regarding the statement "Optimal GeneNet network (Fig. 3B) outperforms the GeneNet networks obtained with most statistical cutoffs (Fig. 2A)" cited by the reviewer, we have now adapted the text for clarity. In the new version, the statement now reads:

"In this case, the optimal GeneNet network yields only a minor improvement over the network obtained with FDR 0.05, which turned out to be the statistical cutoff closest to the optimum."

An example of the difference between FDR-inferred and optimized network is provided for the transcriptomics data in the "Axon guidance pathway" in Figure 7, where these networks are compared to the prior knowledge networks considered.

In Fig. 4, a series of biological reference networks are reconstructed with various levels of error or missing information. During the bootstrapping of the reference network, it is unclear if the incorrect (or missing) information were systematically included (or excluded) in all the runs or if these incorrect/missing data were allowed to be exchanged when generating the adjacency matrices. If exchange is allowed, it is not surprising that the approach can handle incorrect reference information. However, in a real experiment,

we would only have one erroneous biological reference. Here, it may not be possible for this approach to identify the 'correct' result in the presence of 'erroneous' reference. The ability of this approach to reveal new information may thus be limited.

In the incomplete and partial prior knowledge analysis described in Figure 4 we do allow exchange of missing or incorrect edges across the different random references considered. However, we agree with the reviewer that in concrete cases the user has access to a single erroneous biological reference. In order to show what the performance of the approach is in a real world scenario, we have now substantially modified the section describing the application to transcriptomics data to address this point and the lack of validation in general (see Result section "Application to Transcriptomics Data").

Independent validation.

In the new version of the manuscript, we use mutually exclusive training and validation reference networks for the analysis of transcriptomics data. This way, the training reference can also be interpreted as an incomplete version of the true biological reference.

We generated a modified STRING training network by removing CORUM protein complex interactions from it. The resulting network is then tested on CORUM, ensuring that no reference interactions used for training contribute to the validation. Our results show that, the STRING-optimized networks better represent the CORUM complexes than the statistically inferred networks (Fig. 6), hence proving that we can identify correct molecular interactions even in the presence of a single partial (and thus incorrect) prior knowledge. In our opinion this new analysis shows the inference potential of our approach through independent validation.

The glycomics data is a simple data set with ~20 reactants. The concept outlined by the authors needs to be experimentally verified in more complex systems. Otherwise, all that is being done is to exchange one bias for another.

Overall, this is an interesting approach to calculate optimal correlation cut-off values, but its predictive powers require further validation.

In the manuscript, we describe and show how the proposed approach can be applied to metabolomics and transcriptomics data, which are significantly more complex than glycomics data (~1,000 and ~16,000 variables, respectively). As described above, as part of the application to transcriptomics data, we now show that our approach is effective in retrieving biological interactions that are not included in the reference used for optimization, hence proving its predictive potential.

Reviewer #2 (Remarks to the Author):

In this study, Benedetti et al. aimed to develop an algorithm for better correlation-based network inference. Instead of relying on standard statistical cutoffs, their approach selects the cutoff by optimizing the overlap between the inferred and the previously known biological network. It is claimed that the optimal network outperforms networks obtained with statistical cutoffs and is robust to the sample size variation. Overall the paper is well written. The introduction provides good overall coverage of the relevant background literature and description of the problem. However, some sections especially the methodology part needs to be revised/clarified to provide a clearer explanation. Furthermore, it is believed the analysis is quite limited in its scope. These significantly weaken the conclusion of this manuscript and limits its novelty and impact.

Major concerns:

1. There is lack of solid validation of the proposed method. Basically, the study used prior information to select the optimal cutoff based on the overlap between the inferred network edges and the prior network/pathway information (with Fisher's test), but its performance should also be tested in an independent dataset (gold standard).

We agree that independent validation is needed; this comment is in line with a similar criticism by Reviewer #1. We refer to our detailed response above in this response letter (see "Independent validation" on previous page) and we hope that the new version of the manuscript gives a better description of the scope and generalizability of the approach.

2. Even with using the same gold standard for selecting and testing the cutoff, the results based on the proposed algorithm are similar with the statistical cutoff (FDR 0.01 or 0.05, Figures 3 and 5). There is only minor improvement over existing statistical methods when applying the algorithm on the IgG glycomics and metabolomics data analysis. For the transcriptomics data, while FDR 0.05 may yield a much sparse network, the problem can be simply overcome by relaxing the FDR cutoff, eg. FDR 0.1.

The reviewer is correct in pointing out that the improvement provided by our approach over statistical cutoffs, and in particular over FDR 0.05, is limited in the glycomics and metabolomics case. Notably, in the transcriptomics case, our optimization procedure yields major improvements. We have adapted the discussion of those results to provide a clearer description.

While it is true that in principle the user could then choose a more relaxed significance threshold for the statistical cutoff, the fundamental point is that there is no way of knowing *a priori* which statistical method (e.g. FDR or Bonferroni) or significance level (e.g. 0.05 or 0.1) will yield the optimal results. Our approach allows to overcome the arbitrariness of these choices by leveraging the available prior knowledge to determining the "best" (in a biological sense) cutoff automatically. We have edited the manuscript text to make this point, which we believe is at the core of our approach, clearer for the reader. For example, in the Discussion we explicitly address this point as follows:

"Interestingly, for both the glycomics and the metabolomics dataset, the statistical 0.05 FDR cutoff was close to the optimum. However, this performance of FDR is coincidental and cannot be generalized to other datasets or data types. This was corroborated by the analysis of transcriptomics data, for which FDR cutoffs were found to significantly overestimate the optimal cutoffs in many cases. This demonstrates that the performance of FDR cannot be known *a priori* and varies substantially depending on the chosen significance level and data type, and thus does not guarantee an optimal network."

3. The classical Pearson correlation and the partial correlation correspond to completely different biological concepts in the context of network inference. The manuscript doesn't provide sufficient clarification of both measures, instead, it simply compared them and made a conclusion that the partial correlation using GeneNet is superior compared to others. Detailed explanation should be provided on why this is the case.

The reviewer raises a fair point, and we agree that we haven't discussed this properly in the previous version of the manuscript. In the first section of the Results we have added a small paragraph to highlight the conceptual difference between the two measures:

"Note that these two correlation approaches represent substantially different measures of associations: While Pearson correlation quantifies the overall linear association between two variables, partial correlation accounts for the presence of confounders and covariates, therefore removing indirect pairwise associations that only arise due to common mediators. Previous studies have shown that partial correlation selectively represents synthesis steps in metabolic and glycosylation synthesis pathways, while Pearson correlations are widely used, for example, to infer gene co-expression networks. In this paper we considered both these measures for comparison."

4. It should be made clearer how the p-values (before multiple testing correction) were obtained for the Pearson correlation and the partial correlation measure. The different p-value estimation methods also affect the performance.

This is absolutely correct, and we believe that the different behaviors of regular partial (parcor) and GeneNet correlations with sample size is indeed due to the different p-value estimations in the two approaches. We have added a Supplementary Methods file that includes a detailed description of how the p-values are computed for the different correlation measures. We refer to our detailed response to Reviewer #1 above in this response letter (see "Correlation measures" on first page).

Reviewers' Comments:

Reviewer #1:

Remarks to the Author:

I have read the paper again. I looked for the changes to the main document but they were not marked. The response letter to the reviewer suggests the text changes were made to the manuscript but there is no substantial difference from the original submission.

The concern raised previously was that the glycomics data have already been analyzed in a Nature Communication paper published in 2017, and that the added information here is small. The network is well established as glycan diversity on IgG is low.

The figures for metabolic pathway and protein network analysis are interesting, but there is no experimental validation or new biological finding.

Thus, I find that this is a well written paper that has been carefully conducted. The use of Fisher's exact test for model inference is interesting, and indeed using prior experimental data does help focus network inference. However, there is no biological validation in this paper.

Reviewer #2:

Remarks to the Author:

The authors responded some, but not all, of the requested changes and clarifications from the first round of review. It is appreciated that the authors now provided detailed description on how the statistical significance is assessed with Pearson and partial correlations. The scope of the manuscript is also expanded with inclusion of a large scale dataset (transcriptomics dataset with STRING and CORUM as references). However, the section on independent validation using these two references needs further clarification. It is not clear whether all the links between pathway components are identified using the statistical cutoff, or the optimized network cutoff. In addition, since SPRING and CORUM are databases of different types (one is focusing on protein physical interactions, and the other is on protein complex composition), the rationale of choosing these two references as training and testing sets needs to be explained.

Response to the reviewers

Reviewers' comments:

Reviewer #1 (Remarks to the Author):

I have read the paper again. I looked for the changes to the main document but they were not marked. The response letter to the reviewer suggests the text changes were made to the manuscript but there is no substantial difference from the original submission.

We apologize to the reviewer for the inconvenience. The manuscript underwent substantial editing and restructuring from the previous version, and we felt that marking all changes in the main text would have negatively affected readability.

We revised the manuscript to address the lack of validation concern raised in the previous round of reviews. The paper now includes a completely restructured transcriptomics result section where we describe how the application of our approach to transcriptomics data can lead to novel biological findings and outperforms networks inferred with statistical cutoffs.

The concern raised previously was that the glycomics data have already been analyzed in a Nature Communication paper published in 2017, and that the added information here is small. The network is well established as glycan diversity on IgG is low.

The reviewer is correct in stating that the glycomics data reported in this paper have already been analyzed in a previous publication and that the IgG biochemical pathway is well established, also due to the limited diversity of its glycan pool. In fact, this was a desired property for the proof of concept of our approach. By showing how the method works in a “best case scenario”, where we have a well-powered dataset with a well-characterized reference, and by comparing this scenario to sub-optimal scenarios of low sample size or incorrect biological reference, we could provide compelling evidence of the robustness of the overall approach. Only once this had been established could we be confident that the results obtained from more complex datasets, where the prior knowledge is spotty and incomplete (like in the case of metabolomics and transcriptomics) would be meaningful.

The figures for metabolic pathway and protein network analysis are interesting, but there is no experimental validation or new biological finding. Thus, I find that this is a well written paper that has been carefully conducted. The use of Fisher's exact test for model inference is interesting, and indeed using prior experimental data does help focus network inference. However, there is no biological validation in this paper.

Although we agree that we have not performed any new experimental validation experiments, we feel that our analysis of the transcriptomics data provides strong evidence that our approach can be used to infer new biology. We show that a network optimized on one biological reference (in our case the protein-protein interaction network from the STRING database) is able to identify biological interactions not contained in such reference (here protein complexes from the CORUM database). Moreover, it does so significantly better than a network computed with a statistical cutoff, corroborating the overall story of the paper.

Reviewer #2 (Remarks to the Author):

The authors responded some, but not all, of the requested changes and clarifications from the first round of review. It is appreciated that the authors now provided detailed description on how the statistical significance is assessed with Pearson and partial correlations. The scope of the manuscript is also expanded with inclusion of a large scale dataset (transcriptomics dataset with STRING and CORUM as references). However, the section on independent validation using these two references needs further clarification. It is not clear whether all the links between pathway components are identified using the statistical cutoff, or the optimized network cutoff. In addition, since SPRING and CORUM are databases of different types (one is focusing on protein physical interactions, and the other is on protein complex composition), the rationale of choosing these two references as training and testing sets needs to be explained.

We have extensively edited the result section on the transcriptomics data analysis in order to make the rationale, choice and contents of reference databases and findings clearer to the reader. We have also edited Figure 6 labels and caption to avoid confusion between optimized and statistical networks. These edits should address the reviewer's questions and clarify the workflow to the reader.

Regarding the different contents of the two databases used for the analysis, we would like to point out that the field of PPI and protein complexes are deeply interrelated since complexes are made of physically interacting proteins. In fact, STRING, which is one of the most popular PPI databases currently available, contains information about both physical and functional interactions, including protein complex-based interactions¹. Vice versa, the CORUM paper lists protein complexes as one prominent example of functional modules formed by protein-protein interactions². We have now made the relationship between the two databases more evident in the manuscript. Since CORUM is manually curated and thus of potentially higher quality, we relied on this database for the protein complexes information and removed all interactions covered in CORUM from the STRING reference in order to make the two priors completely independent.

See tracked changes in manuscript for details.

Bibliography

1. Szklarczyk, D. *et al.* STRING v11: protein–protein association networks with increased coverage, supporting functional discovery in genome-wide experimental datasets. *Nucleic Acids Res.* **47**, D607 (2019).
2. Giurgiu, M. *et al.* CORUM: the comprehensive resource of mammalian protein complexes—2019. *Nucleic Acids Res.* **47**, D559–D563 (2019).